# Acceptance of Pregnancy-Induced Disease and Intrapersonal Resistance Resources of Pregnant Women—Preliminary Report

**DOI:** 10.3390/ijerph20043199

**Published:** 2023-02-11

**Authors:** Agnieszka Pieczykolan, Ewa Rzońca, Joanna Grzesik-Gąsior, Magdalena Korżyńska-Piętas, Grażyna Iwanowicz-Palus, Agnieszka Bień

**Affiliations:** 1Chair of Obstetrics Development, Faculty of Health Sciences, Medical University of Lublin, 20-081 Lublin, Poland; 2Department of Obstetrics and Gynecology Didactics, Faculty of Health Sciences, Medical University of Warsaw, 00-575 Warsaw, Poland; 3State University of Applied Sciences in Krosno, 38-400 Krosno, Poland

**Keywords:** pregnancy, gestational diabetes, hypertension, acceptance of illness, self-efficacy, health locus of control

## Abstract

The health problems complicating pregnancy are a source of anxiety and concern about the developing fetus’ health and life. The aim of the study was to assess the acceptance of illness and selected intrapersonal resistance resources for women whose pregnancies are complicated by gestational diabetes or pregnancy-induced hypertension and their determinants. The study was conducted from April 2019 to January 2021 in 688 pregnant women who were patients of the pregnancy pathology department and gynecology-obstetrics outpatient clinics in Lublin (Poland), using a diagnostic survey method with the use of the following research tools: Acceptance Illness Scale, Generalized Self-Efficacy Scale, Multidimensional Health Locus of Control Scale, and the standardized interview questionnaire. The study group included 337 women with gestational diabetes and pregnancy-induced hypertension. The control group included 351 women with an uncomplicated course of pregnancy. The level of acceptance of illness in pregnant women with pregnancy-induced diseases is on the border between medium and high acceptance (29.36 ± 7.82). The respondents in the control group had lower levels of self-efficacy (28.47 vs. 29.62) and health locus of control in the internal dimension (24.61 vs. 26.25) (*p* < 0.05). Respondents with pregnancy-induced diseases are characterized by the internal dimension of locus of health control.

## 1. Introduction

Pregnancy is a special moment in a woman’s life, involving the experience of many emotions associated with expecting a child. The mother-to-be prepares for her new role by making changes in her personal and professional life. However, even a normal pregnancy can be considered as a difficult or stressful situation that disrupts the previous state of stability, causing feelings of anxiety, sadness, and a lack of motivation [1,2]. The health problems complicating pregnancy are a source of anxiety and concern about the developing fetus health and life. The occurrence of complications hinders the experience of pregnancy as a joyful time of anticipation of a new life. Moreover, it can determine difficulties in broad self-efficacy and cause psychological problems. There is often a need for hospitalization, which intensifies the sense of anxiety in pregnant woman [3,4].

Gestational diabetes mellitus (GDM) and pregnancy-induced hypertension are one of the most common pregnancy-induced diseases, which can be described as extremely important clinical problems in a woman’s care [5]. The prevalence of gestational diabetes mellitus (GDM) in Europe is estimated to be 15%, and that in Poland is estimated to be 9.7% (data for 2021) [6]. Many women with gestational diabetes experience perinatal complications, including hypertension, urinary tract infections, shoulder dystocia, and the high birth weight of the newborn. Moreover, women with GDM have a high risk of developing type 2 diabetes in the future [7].

The problem of hypertension in pregnancy affects 6–10% of pregnant women worldwide. Pregnant women with hypertension have an increased risk of future cardiovascular diseases (including hypertension or ischemic heart disease) than pregnant women with normal blood pressure [8,9,10].

One of the factors that affect a woman’s approach to a complicated pregnancy is the acceptance of the disease. The level of acceptance of the disease is part of the process of adapting to life with the disease, and it is related to the degree of severity of negative emotional reactions caused by pregnancy-related diseases [11]. Self-efficacy refers to a person’s subjective feeling that he or she is able to bring about a desired situation through the actions taken. In contrast, locus of control of health is associated with participation in preventive programs to maintain health. Both perceptions of self-efficacy and health control are determinants of intentions and actions taken in the health-promoting area, which is of particular importance in women with complicated pregnancies [3,12,13,14].

### Purpose of the Study

The aim of the study was to assess the acceptance of illness and selected intrapersonal resistance resources of women: self-efficacy and the health locus of control by women with pregnancies complicated by gestational diabetes or pregnancy-induced hypertension and the factors determining them. The conducted study may contribute to the recognition of the needs of women with complicated pregnancies, and the obtained results will help in adjusting care to women’s expectations and better understanding their needs, which may improve the quality of perinatal care.

## 2. Materials and Methods

The study was conducted between April 2019 and January 2021 among patients of the pregnancy pathology department (the study group—included all patients of the ward in a given period, who met the criteria for inclusion in the study) of Cardinal Stefan Wyszyński Regional Specialist Hospital in Lublin (the third degree of reference hospital) and patients of the gynecology and obstetrics outpatient clinic (women with uncomplicated pregnancies—control group) of Luxmed Medical Center in Lublin (Poland). Research was suspended twice—at the turn of March and April 2020 and from November to December 2020 due to introduction of lockdown in Poland, related to the emergence of SARS-CoV-2.

Initially, 720 pregnant women hospitalized in the pregnancy pathology department and patients of the gynecology-obstetrics outpatient clinic were included in the study. The selection of the group was purposive. Finally, after verification of the inclusion and exclusion criteria, 688 pregnant women took part in the study. The study group included 337 women, including 181 (53.71%) women with gestational diabetes and 156 (46.29%) women with pregnancy-induced hypertension. The control group included 351 women with an uncomplicated course of pregnancy, confirmed by medical records.

Inclusion criteria for pregnant women in the study were: presence of a comorbid condition in pregnancy (gestational diabetes or pregnancy-induced hypertension), consent to participate in the study, completed 18 years of age (the age of majority in Poland), Caucasian race, native language—Polish, single pregnancy, time from diagnosis of the disease at least 4–5 weeks, use of health care in Poland throughout pregnancy. Exclusion criteria: age under 18, diagnosis of diabetes or hypertension before pregnancy, the presence of gestational diabetes or induced hypertension in a previous pregnancy, family history of diabetes or hypertension (among first-degree relatives), diagnosis of other diseases complicating pregnancy (such as thyroid disease, cancer, liver disease, threatened preterm labor), experienced complications in previous pregnancies, physical, mental, or sensory disabilities.

The diagnosis and classification of gestational diabetes followed the current guidelines of the Polish Society o Gynecologists and Obstetricians and the Polish Diabetes Association when one of the following criteria was met at OGTT: fasting blood glucose between 92–125 mg/dL, 60th-min blood glucose ≥ 180 mg/dL, 120th-min blood glucose between 153–199 mg/dL [15]. On the other hand, the diagnosis of pregnancy-induced hypertension was made according to the recommendations of the Polish Society of Hypertension, the Polish Cardiac Society, and the Polish Society of Gynecologists and Obstetricians, when after 20 weeks of pregnancy: gestational hypertension (with blood pressure ≥ 140/90 mmHg without proteinuria or other biochemical abnormalities) or preeclampsia (with blood pressure ≥ 140/90 mmHg and with the presence of >300 mg of protein in daily urine and/or maternal organ dysfunctions and/or uteroplacental dysfunctions) appears [8,16].

Criteria for the inclusion of pregnant women in the control group: uncomplicated course of pregnancy confirmed on the basis of medical records—Pregnancy Card, completed 18 years of age, consent to participate in the study, native language—Polish, single pregnancy, use of health care in Poland throughout the pregnancy.

Pregnancy medical records card is the basic document of pregnant women in Poland, which is set up by the person who provides care (doctor, midwife) during the first visit and completed manually during subsequent visits. Pregnancy medical records card, besides basic data—name, date of birth, home address, date of last menstrual period, and expected date of delivery, also contains information on the pregnant woman’s condition—blood group and RH factor, week of pregnancy, woman’s weight, blood pressure, height of uterine fundus, possible edema, varicose veins, or other ailments. The results of laboratory tests, including blood count, urinalysis, TSH, the result of an oral glucose tolerance test, and the presence of toxoplasmosis, rubella, or HIV antibodies are also completed periodically. At each visit, the fetal heart rate and other parameters obtained during the ultrasound examination are also recorded. The maternity notes are considered a reliable source of information on the condition of the pregnant woman and the developing fetus since the entries are made on the basis of the original test results from the laboratory. Moreover, each entry must be confirmed by the stamp and signature of the person who made it.

The study used a diagnostic survey with questionnaires. The following instruments were applied: the Acceptance of Illness Scale (AIS), the Generalized Self-Efficacy Scale (GSES), the Multidimensional Health Locus of Control Scale (MHLC), and a standardized interview questionnaire with items concerning the participants’ characteristics.

AIS (Felton et al., adapted to Polish conditions by Juczyński), is used to assess the degree of acceptance of illness by adults. The tool consists of 8 statements representing the negative consequences of poor health status, which the individual assesses based on a 5-point scale, where 1 means strongly agree and 5 means strongly disagree. The overall score is in the range of 8–40; a low score indicates a lack of acceptance as well as adjustment to the disease, while a high score indicates acceptance of the disease by respondent. Cronbach’s internal consistency coefficient α is 0.85, and the reliability of the Polish version of the AIS is similar to the original version; Cronbach’s α is 0.82 [17].

GSES evaluates an individual’s value system manifested in their ability to cope with difficult situations. The scale comprises 10 statements rated on a scale of 1 to 4 (1—disagree, 2—somewhat disagree, 3—somewhat agree, 4—agree). The total score reflects the overall level of self-efficacy, with higher scores indicating more self-efficacy. Scores between 10 and 24 points are interpreted as a low level of self-efficacy, 25–29 points—moderate, and 30–40 points—a high level of self-efficacy. Cronbach’s alpha for scale reliability is 0.85, and the internal consistency of the GSES ranges between 0.76 and 0.91 [18].

MHLC comprises 18 statements rated on a six-item scale, which represent convictions referring to generalized expectations in three health locus of control dimensions: internal factors (I am in control of my health), external factors/impact of others (my health results from the actions of others, including medical personnel), and the belief that one’s health results from random events. The total score for each subscale is between 6 and 36 points. Higher scores indicate a stronger belief that the factor of interest affects one’s health. Scores are interpreted based on the median value: those above the median are considered high, and those below the median are considered low. Scale reliability is 0.64 for internal control, 0.59 for impact of others, and 0.63 for random events [19].

The study was approved by the Lublin Medical University Bioethics Committee (approval no. KE-0254/166/2018). Respondents were informed that participation was voluntary, and that study results were anonymous and to be used exclusively for research purposes.

### Statistical Analysis

The obtained results of the study were collected and statistically analyzed using STATISTICA 13.3. The values of the analyzed non-measurable parameters (age, marital status, education, perceived family wealth, planned pregnancy, BMI, number of pregnancies, week of pregnancy) were presented using the number and percentage.

In the descriptive analysis, data were presented using mean value, standard deviation, frequency, and percentage. To examine differences in parameters without a normal distribution appropriate statistical tests were used (Mann–Whitney U test, Kruskal-Wallis test). Stepwise regression was used to identify predictors of AIS, GSES, and MHLC scores. Stepwise regression is a method of fitting the regression models in which the choice of predictive variables is performed using an automatic procedure [20].

The correlation between quantitative variables was calculated. Pearson’s r-Pearson correlation was used to test the relationship between the selected variables, while rho-Spearman correlation was used for ordinal variables. The analysis considered a continuous variable (age, education, BMI, week of pregnancy) and a categorical variable (marital status, perceived family wealth, planned pregnancy). Two categories were created for the categorical variable. Multiple linear regression analysis with moderating factors was performed to evaluate the variables (F-test). The study assumed a significance level of *p* < 0.05.

## 3. Results

Table 1 shows the characteristics of respondents. In total, 688 women took part in the study; the study group (women with an abnormal course of pregnancy) consisted of 337 people (49.0%), while 351 people (51.0%) were the control group, i.e., pregnant women with an uncomplicated course of pregnancy. In the study group, the majority of respondents were women under 35 years of age (88.4%), married (66.2%), had a university degree (67.4%), assessed their family’s wealth as average (70.9%), stated that the current pregnancy was planned (62.6%), had a BMI indicating normal weight (43.0%), were pregnant for the first time (52.5%), and were between the 31st and 36th weeks of pregnancy (36.5%). In the control group, the majority of women were under 35 years of age (92.0%), married (67.5%), had a university degree (74.9%), rated their family wealth as average (73.5%), stated that the current pregnancy was planned (72.4%), had a normal BMI score (42.2%), were pregnant for the first time (57.0%), and were between the 31st and 36th weeks of pregnancy (45.0%).

The mean AIS score in the group of women with complicated pregnancies was 29.36 ± 7.82 (Me = 31.00). Pregnant women with an abnormal course of pregnancy had a lower level of the GSES (28.47 vs. 29.62) and a lower level of the locus of health control in the internal dimension (IHLC) (24.61 vs. 26.25) compared to pregnant women with a normal course of pregnancy. The indicated correlations were statistically significant (*p* < 0.05)—Table 2.

Table 3 shows the regression analysis for the acceptance of illness, the generalized sense of self-efficacy, and the health locus of control in the study group. The regression analysis showed that explanatory variables with a significant effect on the AIS mean value were age (β = 0.245; *p* = 0.043) and marital status (β = 4.676; *p* = 0.041). It was shown that the acceptance of the disease increased with age and was positively related to being in a relationship. The F-test results were statistically significant, with the estimated model explaining 6.9% of the total variance of the outcome (F = 2.061; *p* = 0.028). Statistically significant predictors for the model of the self-efficacy variable (GSES) were age (β = 0.132; *p* = 0.048), wealth (β = −0.120; *p* = 0.039), and planning a pregnancy (β = 0.159; *p* = 0.016). It was shown that self-efficacy increased with age and was positively associated with a better economic situation and the fact of planning the current pregnancy. The F-test results were statistically significant, with the estimated model explaining 12.5% of the total variance of the outcome (F = 3.945; *p* < 0.001). Multilevel variable scanning showed that the location of health control in terms of chance influence (CHLC) was associated with lower age (β = −0.166; *p* = 0.014) and lower education of pregnant women (β = −0.221; *p* = 0.001). The analysis of variance is statistically significant in the present case (F = 3.474; *p* < 0.001). The model explains 11% of the variance.

The statistical analysis showed significant positive correlations between acceptance of illness and self-efficacy (r = 0.233), as well as between the location of health control in the dimension of influence of others and health control in the dimensions of internal influence (r = 0.229) and influence of chance (r = 0.314). There was no correlation between the relationship between the other variables (*p* > 0.05)—Figure 1, Figure 2 and Figure 3.

## 4. Discussion

Pregnancy is a special period in a woman’s life; it changes their life goals and hierarchy of values and forces new behaviors and new expectations. Even a normal pregnancy can be considered a difficult or stressful, disruptive situation. Even the highest quality preconception care and preparation for pregnancy cannot eliminate the possibility of various complications or diseases that may occur. Diseases that complicate pregnancy are associated with an increased risk of complications for the mother and/or child. Moreover, a woman may feel guilty about an existing condition, which negatively affects the perinatal period [1,4,5,21,22]. Diabetes mellitus and pregnancy-induced hypertension are the most common diseases that complicate pregnancy, so the respondents of the study were women who were diagnosed with these diseases during pregnancy [5].

The way of coping with an emerging disease varies depending on a woman’s experience. Putting oneself in the role of the patient is associated with acceptance of the disease. According to the literature, the degree of acceptance of the disease does not depend on the disease entity itself, but on the timing of its onset—people with a sudden onset of the disease have a higher degree of acceptance of the disease compared to patients with chronic conditions. Acceptance of the disease by pregnant women is more difficult, as it applies to the mother and child’s health and life. A higher level of acceptance of one’s condition is associated with an increased level of motivation of individuals in overcoming the limitations of the existing condition and a sense of independence from others [23,24]. In the study group, the average level of acceptance of illness was 29.36, which is a borderline result between medium and high. The result may be due to the short time since diagnosis and respondents’ strong desire to cope with difficulties. These results are consistent with those of other authors analyzing the level of acceptance of the disease by pregnant women. In a study by Bien et al. who studied pregnant women with GDM, the acceptance of the disease was 30.66, while in a study by Iwanowicz-Palus et al., the level of acceptance of the disease among pregnant women with hyperglycemia was higher, at 31.37 [24,25]. A high level of acceptance of illness signifies the patient’s adaptation to the disease and its adverse consequences. People who accept the disease better understand the course of it, have an optimistic approach to their illness, show confidence in doctors and therapeutic methods that are used, and actively participate in the therapeutic process [11].

Respondents with low economic status presented the lowest level of awareness of the disease, which may be due to more difficult access to health care [26]. The results of a study by Rogon et al. among patients with type 2 diabetes indicated that acceptance of the disease is often higher in younger people than in older ones, which may be due to the greater overall functional capacity of the body in daily life [27]. Analysis of the results of our study showed that illness acceptance increased with age and was positively associated with being in a relationship. Being in a relationship may improve better coping with the disease. For example, by helping and supporting a loved one with overcoming the problems associated with the illness and adapting to the limitations of the disease [28], which may also justify our results.

Self-efficacy was another of the assessed determinants of coping with an abnormal course of pregnancy. In our study, the average level of self-efficacy in the group of pregnant women with an abnormal course of pregnancy was 28.47, which was in the upper limits of the average reference values; it was statistically lower compared to the control group—pregnant women with an uncomplicated course of pregnancy, in which it was 29.62. The above correlation allows us to assume that women with a complicated pregnancy may blame themselves for the occurrence of the illness- for not taking enough care of themselves/for poor prophylaxis of the disease. An analysis of the literature showed a limited number of articles assessing the impact of self-efficacy on the lives of women with abnormal pregnancies. A study conducted by Mahmoodabad et al. among pregnant Iranian women with a normal course of pregnancy found that the overall self-efficacy score was 23.31, lower than in our study [29]. Values similar to those obtained in our own study were achieved in primiparous women in the third trimester of pregnancy (28.29) [30], in obese pregnant women with threatened preterm labor (28.02) [31], in pregnant patients with hyperglycemia (31.58) [25] and in women after miscarriage (30.29) [32]. The regression analysis of our study showed that the explanatory variables for self-efficacy were age, wealth, and planning a pregnancy, which may be due to the fact that older women with higher socioeconomic status and those planning a pregnancy were more aware of pregnancy and the need to take care of their own health and that of their unborn child. In contrast, the results of a study by Vance et al. showed that low self-efficacy in mothers was influenced by factors such as lower family wealth, being married, and living together with a partner [33].

The presence of a disease that complicates the course of pregnancy forces pregnant women to lead a healthy lifestyle. A high level of self-confidence and self-efficacy, as well as adequate knowledge about the condition and its impact on the pregnancy and the baby, increases a woman’s motivation to make health behavior changes for the better [34]. Harrison et al. in their study showed that pregnant women with GDM expect medical staff to provide reliable and specific information about physical activity during pregnancy, which translates into increased levels of self-efficacy in taking care of their own and their baby’s health [35].

In our study, we also analyzed the locus of health control among pregnant women, which is important in the context of coping with difficult situations such as a high-risk pregnancy. The results indicate that of all dimensions of health locus of control, the highest values were obtained in the scale of internal control. In the study, the average level of health control in the study group in the dimension of internal influence was 24.61 and was statistically lower compared to the control group, in which the level was 26.25. In the study group, the level of health control in the dimension of influence of others was 22.15 and in the dimension of influence of chance was 20.36, while in the control group, the level of health control in the dimension of influence of others was 20.99 and that in the dimension of influence of chance was 19.24. The obtained differences were not statistically significant. This trend is, however, a desirable phenomenon in pregnant women, as the conviction of deciding on one’s own health will translate into taking health-promoting behaviors, such as following a low-carbohydrate diet, performing control tests recommended by the doctor, and systematically measuring blood glucose and blood pressure, which will reduce the risk of future various complications in pregnancy [35,36]. Individuals with a high level of internal health control are more likely to exhibit health-promoting behaviors compared to those with a low sense of health control. However, a high level of health control in the chance impact dimension reduces the likelihood of adherence to medical recommendations and health-promoting behaviors [37].

External locus of control is associated with, among other things, a higher incidence of depression in low-income individuals and young people who were exposed to financial difficulties in childhood. In contrast, a more internal health locus of control may protect or mitigate the impact of the stress of financial hardship on the mental health of these individuals [38,39,40,41]. However, Frankham et al. showed that internal and economic loci of control, however, are related to financial hardship (both subjective and objective) [42].

Szczepinska’s study on the relationship between the health locus of control and the expectations of pregnant women receiving care at the Pregnancy Pathology Clinic showed that those with higher education, as well as with very good and good material conditions, presented the highest internal health locus of control, while health control in the dimension of influence of others and influence of chance presented lower [43]. In our study, the regression analysis showed that age and education were explanatory variables for health control in the dimension of the influence of chance. It can be concluded that younger people with lower levels of education were not fully aware of the impact of their actions on health, counting on the positive impact of chance situations.

There are few scientific reports in the literature addressing the issues of acceptance of illness and psychosocial resilience resources of pregnant women: self-efficacy and health locus of control and their determinants, especially in the case of the occurrence of diseases complicating pregnancy, which is important in the care of women with abnormal pregnancy [25,29,30,31,32,42,43]. The implemented care should be based on the real needs, capabilities, and expectations of pregnant women to cope with the situation. Proper, individualized care provided by medical staff, education on nutrition, physical activity and self-control, support, and a holistic approach to pregnant women with a complicated course of pregnancy can optimize midwifery care and positively affect the psycho-physical condition of pregnant women with an abnormal course of pregnancy.

One of the strengths of the study is the complementary approach to intrapersonal resilience resources of pregnant women in the presence of selected diseases complicating pregnancy. It should be noted that the inclusion criteria for women in the study and control groups were carefully defined. Moreover, the criteria for the diagnosis of comorbidities in pregnancy, gestational diabetes and pregnancy-induced hypertension, were defined in detail. An undoubted advantage of the study was direct contact with all respondents, personal data collected on the basis of pregnant women’s medical documentation, and the use of standardized research tools therefore in the future the results of our study can be compared with these obtained by other researchers who can continue the study and draw conclusions.

The study is not without limitations. Among the weaknesses of the study, we can distinguish: the limited sample size, the fact that the results of the conducted study represent the studied group of pregnant women with gestational diabetes and pregnancy-induced hypertension, who were cared for in only one city, although, on the other hand, they were residents of a large voivodeship city in Poland. The limitations of the above study are also due to the research method used and the factors influencing pregnant women’s responses, such as their current emotional state or hospital stay. Therefore, it is advisable, to confirm the correlations obtained in studies among pregnant residents of other cities and even countries. In the future, it would also be advisable to detail the sociodemographic variables of the respondents.

## 5. Conclusions

The level of acceptance of the illness in pregnant women with pregnancy-induced diseases is on the border between medium and high acceptance. Higher age and being in a relationship are the independent factors that increase the acceptance of the illness by pregnant women with pregnancy-induced diseases.

The level of self-efficacy is positively correlated with the level of acceptance of the illness—the higher the level of self-efficacy, the higher the level of acceptance of the disease presented by pregnant women with pregnancy-induced diseases.

The self-efficacy of pregnant women with pregnancy-induced diseases is at an average level, and the factors influencing this self-efficacy are age, assessment of family wealth, and family planning.

The studied pregnant women with complicated pregnancies are characterized by the internal dimension of the location of health control. Factors influencing health control in the dimension of chance impact are age and education—health control in the dimension of chance impact increases with age and is positively related to higher education.

## Figures and Tables

**Figure 1 ijerph-20-03199-f001:**
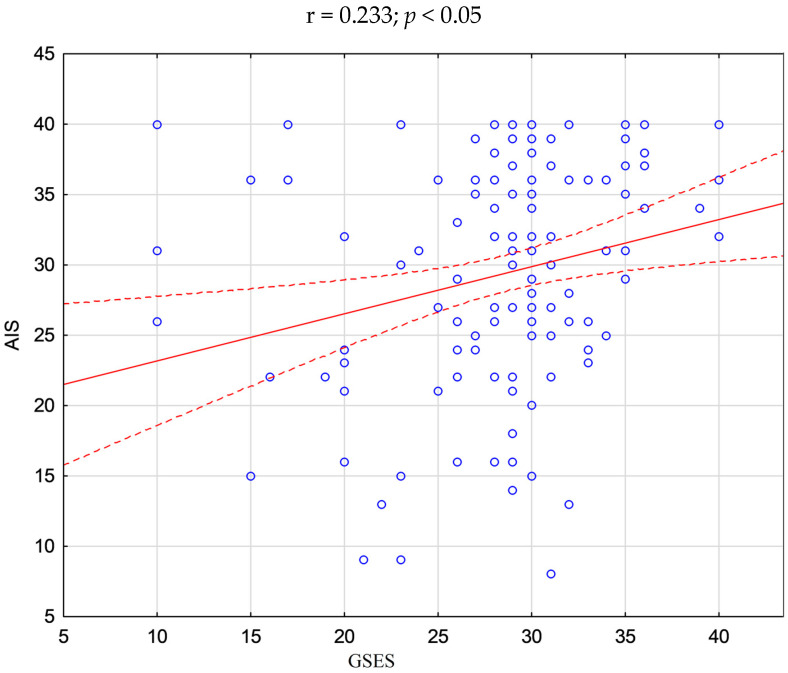
Scatter plot—correlation between AIS and GSES in the study group of pregnant women.

**Figure 2 ijerph-20-03199-f002:**
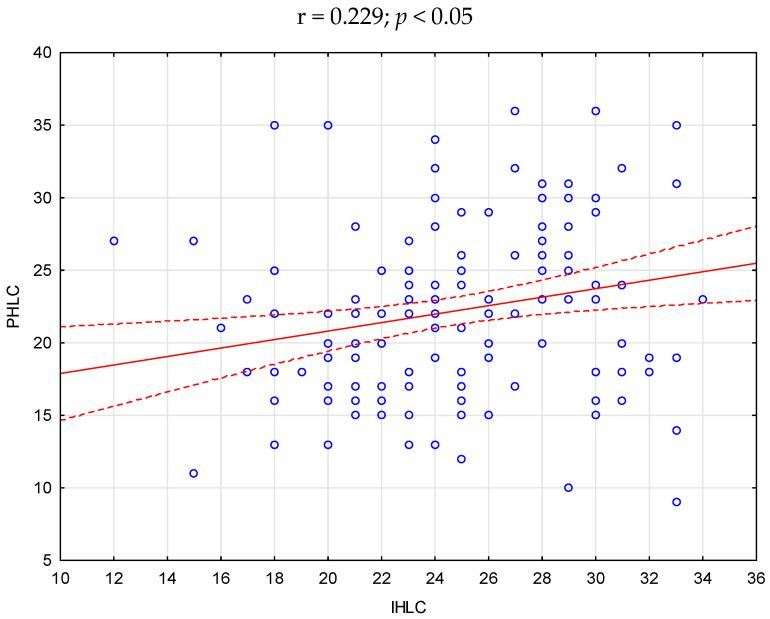
Scatter plot—correlation between PHLC (health control in the dimension of influence of others-powerful others) and IHLC in the study group of pregnant women.

**Figure 3 ijerph-20-03199-f003:**
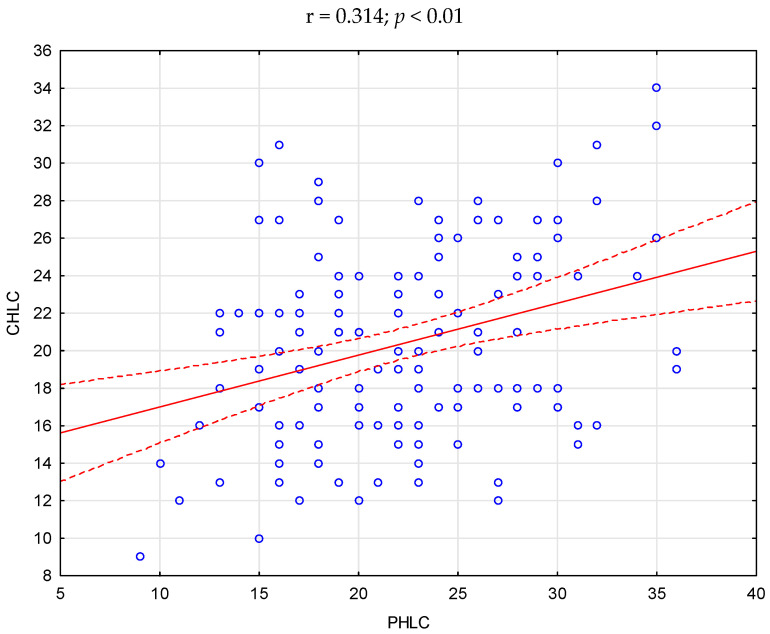
Scatter plot—correlation between CHLC and PHLC in the study group of pregnant women.

**Table 1 ijerph-20-03199-t001:** Socio-demographic characteristics of women in the study.

Characteristics of the Group		Case Group		Control Group		Statistical Analysis
		*n*	%	*n*	%	
		337 (49.0%)		351 (51.0%)		---
Age	≤35 years old	298	88.4	323	92.0	*p* = 0.111
	>35 years old	39	11.6	28	8.0	
Marital status	Single	13	3.9	21	6.0	*p* = 0.042
	Married	223	66.2	237	67.5	
	In a relationship	101	29.9	93	26.5	
Education	Other than higher	110	32.6	88	25.1	*p* = 0.028
	University degree	227	67.4	263	74.9	
Perceived family wealth	Very rich/rather rich	86	25.5	74	21.1	*p* = 0.072
	Average	239	70.9	258	73.5	
	Rather poor	12	3.6	19	5.4	
Planned pregnancy	Yes	211	62.6	254	72.4	*p* = 0.006
	No	126	37.4	97	27.6	
BMI	Correct	145	43.0	148	42.2	*p* = 0.285
	Overweight	118	35.0	140	39.9	
	Obese	74	22.0	63	17.9	
Number of pregnancies	1	177	52.5	200	57.0	*p* = 0.339
	2	81	24.0	84	23.9	
	≥3	79	23.5	67	19.1	
Week of pregnancy	≤26 weeks	64	19.0	51	14.5	*p* = 0.0057
	27–30 weeks	115	34.1	89	25.4	
	31–36 weeks	123	36.5	158	45.0	
	≥37 weeks	35	10.4	53	15.1	

**Table 2 ijerph-20-03199-t002:** Acceptance of Illness, Generalized Self-Efficacy and Multidimensional Health Locus of Control.

Variables		Study Group					Control Group				
		M	SD	Me	Min	Max	M	SD	Me	Min	Max
AIS		29.36	7.82	31.00	8.00	40.00	---	---	---	---	---
GSES		28.47	5.44	29.00	10.00	40.00	29.62	5.22	30.00	8.00	40.00
		Z = 2.081; *p* = 0.037									
MHLC	Internal	24.61	4.57	24.00	12.00	34.00	26.25	4.42	27.00	12.00	35.00
		Z = 3.116; *p* = 0.002									
	Impact of others	22.15	5.82	22.00	9.00	36.00	20.99	5.96	22.00	6.00	36.00
		Z = −1.350; *p* = 0.177									
	Random events	20.36	5.14	20.00	9.00	34.00	19.24	6.21	19.00	6.00	34.00
		Z = −1.777; *p* = 0.076									

AIS—Acceptance of Illness; GSES—Generalized Self-Efficacy Scale; MHLC—Multidimensional Health Locus of Control Scale.

**Table 3 ijerph-20-03199-t003:** Regression analysis results for AIS, GSES, and CHLC in the study group.

Predictors	AIS F = 2.061; *p* = 0.028; R^2^ = 0.069				
	*B*	*SE*	*β*	*t*	*p*
Age	0.245	0.120	0.140	2.034	0.043
Marital status (in a relationship)	4.676	2.285	0.123	2.049	0.041
Education	−0.128	1.186	−0.007	−0.108	0.914
Perceived family wealth (average)	0.385	0.783	0.029	0.491	0.624
Planned pregnancy (yes)	1.680	1.180	0.096	1.424	0.156
BMI	−0.046	0.103	−0.028	−0.449	0.654
Week of pregnancy	−0.192	0.117	−0.098	−1.639	0.102
Predictors	GSES F = 3.945; *p* < 0.001; R^2^ = 0.125				
	*B*	*SE*	*β*	*t*	*p*
Age	0.152	0.077	0.132	1.987	0.048
Marital status (in a relationship)	−0.228	1.450	−0.009	−0.157	0.875
Education	1.473	0.754	0.125	1.955	0.052
Perceived family wealth (average)	1.032	0.498	−0.120	2.075	0.039
Planned pregnancy (yes)	1.820	0.750	0.159	2.428	0.016
BMI	−0.052	0.065	−0.049	−0.791	0.430
Week of pregnancy	−0.006	0.075	−0.005	−0.084	0.933
Predictors	CHLC F = 3.474; *p* < 0.001; R^2^ = 0.111				
	*B*	*SE*	*β*	*t*	*p*
Age	−0.204	0.083	−0.166	−2.468	0.014
Marital status (in a relationship)	1.622	1.568	0.061	1.034	0.302
Education	−2.802	0.815	−0.221	−3.436	0.001
Perceived family wealth (average)	−0.154	0.538	−0.017	−0.285	0.776
Planned pregnancy (yes)	−0.458	0.811	−0.037	−0.565	0.573
BMI	−0.025	0.071	−0.022	−0.357	0.721
Week of pregnancy	0.115	0.081	0.083	1.426	0.155

AIS—Acceptance of Illness; GSES—Generalized Self-Efficacy Scale; CHLC—Multidimensional Health Locus of Control Scale (chance health locus of control); Test–Revised: β—standardized coefficients. SE—bootstrapped standard errors.

## Data Availability

The data presented in this study are available on request from the corresponding author.

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
