# Peer review of "Acceptance of Pregnancy-Induced Disease and Intrapersonal Resistance Resources of Pregnant Women—Preliminary Report"

_ijerph, 2023, doi:10.3390/ijerph20043199_

Round 1

Reviewer 1 Report (Previous Reviewer 3)

The study of Pieczykolan, et al, evaluated the acceptance of the disease, the self-efficacy and the assessment of health control in a sample of 688 women with pregnancies complicated by gestational diabetes or pregnancy-induced hypertension, also investigating the factors determining them. This topic is relevant and interesting, since a high level of acceptance illness means the patient’s adaptation to the disease and its adverse consequences; indeed, people who accept the disease better understand the course of it, have an optimistic approach to their illness, show confidence in doctors and therapeutic methods that are used and actively participate in the therapeutic process.

From the last round of revisions, the manuscript has been improved; several details are better explained and clearer, i.e., the inclusion and exclusion criteria, the temporal period and the geographical catchment area, the discussion section and the limits and strengths of the study.

Despite that, following, you can find my concerns and suggestions along the manuscript, in a very schematic structure.

Materials and Methods

·         Line 92: please provide a brief explanation of why only people speaking Polish were included in the study.

·         Line 94: in order to minimize potential misclassification errors and be more sure that these diagnosis of diabetes and hypertension are more likely induced by pregnancy, I would add the complete list of risk factors to the exclusion criteria. Such as, for example, family history of diabetes (first degree relative with the condition?)

·         The statistical analyses are not too complicated, but they are reported and described clearly and linearly. The descriptive analyses are very concise, please provide more details (maybe a list) regarding the variables considered as covariates in the model; and reported in Table 1.

·         Lines 172-173: perhaps, I would better explain the stepwise regression method or, at least, I would cite a supporting article. An article or reference that supports the various statistical analyses chosen, applies to each step of this paragraph.

·         Line 177: is clear that the outcome of interest is, separately, one of the three scales used (AIS, GSES, and CHLC. But, how it was considered in the statistical model estimated? Was the outcome considered as a continuous variable or as a categorical variable? And if considered as a categorical variable, which was the number of categories created?

If the outcome is continuous then a linear regression model is fine; otherwise not (multinomial or logistic model would be more appropriate). Please specify.

·         Table 1: given the clinical question, it would also be interesting to add "widow" as a category for Marital status. As long as you have the information, obviously. Regarding the variable Education, why only these two categories were considered? In the “secondary” category, were included also patients without any education?

·         Lines 210-211 / 216-217: please correct the sentences; it is incorrect state that the F-test represents the x% of the variance. To be clearer, the estimated model explains the x% of the total variability of the outcome. Maybe you could write something like this: “The F-test results statistically significant; with the estimated model explaining the x% of the total variance of the outcome”.

·         Line 220: the sentence “the adopted model is a good fit to the data” is partially incorrect. Based on what? All right, the model was statistically significant (p-value < .001), but the R-square values are too low to state that the model is a good fit to the data; furthermore, we don’t have an official threshold for the R-square allowing us to say something like this. Be less enthusiastic in this comment.

Discussion

·         The addition of the strengths and limitations is fine to me.

Author Response

We wish to thank you very much for the time you have taken to read and review our paper, as well as for your feedback and suggestions for improvement. We hope the changes we have made improve the overall quality of the paper in line with your expectations.

Materials and Methods

  • Line 92: please provide a brief explanation of why only people speaking Polish were included in the study.

We would like to emphasize that our study was carried out in Poland, and the research tools that we used were validated and translated into Polish. We wanted the respondents to understand the instructions and questions well, and at the same time be able to give reliable answers.

  • Line 94: in order to minimize potential misclassification errors and be more sure that these diagnosis of diabetes and hypertension are more likely induced by pregnancy, I would add the complete list of risk factors to the exclusion criteria. Such as, for example, family history of diabetes (first degree relative with the condition?)

We have completed the exclusion criteria because we really wanted the study to be conducted in pregnant women whose onset of the disease was due to pregnancy.

  • The statistical analyses are not too complicated, but they are reported and described clearly and linearly. The descriptive analyses are very concise, please provide more details (maybe a list) regarding the variables considered as covariates in the model; and reported in Table 1.

We have completed the description of the variables presented in Table 1 in the "Statistical analysis" section.

  • Lines 172-173: perhaps, I would better explain the stepwise regression method or, at least, I would cite a supporting article. An article or reference that supports the various statistical analyses chosen, applies to each step of this paragraph.

We have added literature to this section of the article

  • Line 177: is clear that the outcome of interest is, separately, one of the three scales used (AIS, GSES, and CHLC. But, how it was considered in the statistical model estimated? Was the outcome considered as a continuous variable or as a categorical variable? And if considered as a categorical variable, which was the number of categories created?

Continuous variable and categorical variable were taken into account in the analysis - the information was  complemented in the article

If the outcome is continuous then a linear regression model is fine; otherwise not (multinomial or logistic model would be more appropriate). Please specify.

We added the information which model of the regression was used in the description of the statistical analysis

  • Table 1: given the clinical question, it would also be interesting to add "widow" as a category for Marital status. As long as you have the information, obviously.

Thank you for the suggestion, however, there were no widows, divorcees among the respondents, so they were not included

Regarding the variable Education, why only these two categories were considered? In the “secondary” category, were included also patients without any education?

Thank you for your comment, in the Education variable should be: other than higher education, where we qualified respondents with secondary and vocational education. There were no women in any of the surveyed groups without any education.

  • Lines 210-211 / 216-217: please correct the sentences; it is incorrect state that the F-test represents the x% of the variance. To be clearer, the estimated model explains the x% of the total variability of the outcome. Maybe you could write something like this: “The F-test results statistically significant; with the estimated model explaining the x% of the total variance of the outcome”.

As rightly suggested, the sentences have been changed.

  • Line 220: the sentence “the adopted model is a good fit to the data” is partially incorrect. Based on what? All right, the model was statistically significant (p-value < .001), but the R-square values are too low to state that the model is a good fit to the data; furthermore, we don’t have an official threshold for the R-square allowing us to say something like this. Be less enthusiastic in this comment.

The notation of this sentence have been changed.

Discussion

  • The addition of the strengths and limitations is fine to me.

Thank you for all your comments and suggestions, we hope we have answered all of them reliably and as you  expected.

Reviewer 2 Report (Previous Reviewer 2)

Thank you for you correction. 

Author Response

We wish to thank you very much for the time you have taken to read and review our paper, as well as for your feedback and suggestions for improvement.

This manuscript is a resubmission of an earlier submission. The following is a list of the peer review reports and author responses from that submission.

Round 1

Reviewer 1 Report

This very interesting study analyzes the acceptance of illness, self-efficacy, and health control by women with pregnancy-induced diseases. This topic is very important and the study significantly contributes to the field. Overall the study is well-designed and the conclusions are supported by the results. My concerns are related to the selection of the study group. Previous pregnancies and especially complications during previous pregnancies might significantly affect the attitude towards the current disease, so the authors should more precisely describe the study group: were all cases primiparous, and if multiparous pregnancies were included, how many of these women experienced complications in previous pregnancies? If such cases were enrolled, then additional analysis within this subgroup might improve the study design. 

Reviewer 2 Report

In the manuscript Acceptance of illness, self-efficacy and health control by women with pregnancy-induced diseases: gestational diabetes, pregnancy-induced hypertension “ the authors showed that The level of acceptance of illness in pregnant women with pregnancy-induced diseases is on the border between medium and high acceptance. The respondents in the control group had lower levels of self-efficacy and health locus of control in the internal dimension (p < 0.05). Respondents with pregnancy-induced diseases are characterized by the internal dimension of locus of health control.

The author has presented some basic findings. The study is poorly designed and is only a descriptive survey.  Previous studies (also in Lublin) reported similar results. I cannot find any novelty in this study.

The effect of other factors such as: effects of stress hormone, hereditary hypertension, effects of first time, second, etc., pregnancy.

How do you determine sample size?

Reviewer 3 Report

Manuscript: IJERPH-2033157

Title: Acceptance of illness, self-efficacy and health control by women with pregnancy-induced diseases: gestational diabetes, pregnancy-induced hypertension

Comments:

The study of Pieczykolan, et al, evaluated the acceptance of the disease, the self-efficacy and the assessment of health control in a sample of 688 women with pregnancies complicated by gestational diabetes or pregnancy-induced hypertension, also investigating the factors determining them. This topic is relevant and interesting, since a high level of acceptance illness means the patient’s adaptation to the disease and its adverse consequences; indeed, people who accept the disease better understand the course of it, have an optimistic approach to their illness, show confidence in doctors and therapeutic methods that are used and actively participate in the therapeutic process.

Despite that, and the interesting results, the study has important flaws and limits.

Following, you can find my concerns and suggestions along the manuscript, in a very schematic structure.

Abstract

·       The abstract is lacking any numeric results. Please provide some. Furthermore, is reported the years of evaluation or enrolment of the cohort of women, but it is not reported where the sample was recruited. In addition, the Authors write about (in one of the last sentences) the “control group”, but from the Abstract is not clear what is the control group. It was not defined in the sentences before and is not possible to understand which was the control group. Please, although in the Abstract, better describe the methods used in the study.

Introduction

·       Line 37: find a synonymous of the word “balance”, I think that “balance” is not the appropriate term.

·       Line 58:  change “caused by the diseases” in “caused by the pregnancy related diseases”.

·       When at line 65 the Authors state the aim of the study, please state also the implications and the relevance of this aim for the public health (since these results can play an important role in addressing policies of social assistance to pregnant women with pregnancy related diseases).

Materials and Methods

·       Authors state, at Lines 72-74, that the study was conducted in a outpatient clinic in Lublin (Poland); please provide more details on the catchment area where the study was performed and the sample enrolled. How many patients are taken in care by this clinic usually? How big, and populated, is the catchment area that this outpatient covers?

·       Line 77, Authors cite “medical records” used to confirm complications or comorbidities during the pregnancy of the 688 women enrolled in the study. Please provide more details on these “medical records”. Are these records filled with information collected by the clinic, or by the NHS? Which kind of information is collected in these records? In addition to these needed and fundamental specifications, please discuss also the limitations that the use of this (limited, I guess) source of data can imply and the bias that could introduce in the study.

·       Figure 1; 720 women were initially enrolled in the study, and exclusion criteria were applied. All clear. But these 720 women how they were enrolled in the study? Were all taken in care by the outpatient clinic in the study period?

·       The evaluation scales are well and clearly defined. Only one suggestion, since you already defined the acronym of each of the three scales (AIS, GSES, MHLC), during the following sentences in the manuscript, please do not write always the full name followed by the acronym,, it could make the manuscript more “light”.

Materials and Methods

The statistical analyses must be revised.

·       Line 145: “appropriate statistical tests”, please, which tests? Please describe and list the test used.

·       The sentences describing the stepwise regression are not linear and clear, please rephrase them.

·       The regression analysis, which regression analysis was? Linear? Please specify.

·       To test the difference of the means of AIS, GSES and MHLC assessments between the two groups of pregnant women, you must use a T-test for independent samples, and not the correlation as written. This test is the more suitable when the aim is to determine whether there is statistical evidence that the associated population means are significantly different, between two groups.

Results

·       Table 1, please round at the first decimal cypher. Furthermore, reporting only the p-value and not the value of the statistical test could simplify the reading of the Table.

·       Table 1, please control the sums of the values reported in Table. Some percentages do not sum to 100%.

·       Line 181; rephrase the sentence “that explanatory variables for the AIS variable were…” with something like “that explanatory variables with a significant effect on the AIS mean value were…”

·       Table 3; since most of the variables inserted in the linear (I guess) regression model were categorical or ordinal, could it be useful insert in the model, for each categorical variable, dummy variable, ne for each category; this would make the results more easy to be interpreted.

·       Line 184; “the model presented was statistically significant” is not properly correct by a statistical point of view. The F-test was significant, as the Authors correctly reported.
Test F is used to test whether there is a significant relationship between the dependent variable and the set of independent variables, i.e., the entire multiple regression model. Since the Test resulted significant, you can state that there is a linear association between the dependent variable and at least one independent variable. Be more cautious in the interpretation.

Discussion

·       When specifying details on the sample size of the study, please insert also a consideration regarding the power of the study guaranteed from the actual sample size.

·       No limitations were discussed in the Discussion section. Instead, the limitations of this study are several, and of relevant importance (the data source used, the limited sample size, the selection bias in the enrolment of the sample, etc). Please consider them in the Discussion, to give to the reader a more comprehensive vision of the results, and better read them.